# Evaluation of Pilot-Scale Radio Frequency Heating Uniformity for Beef Sausage Pasteurization Process

**DOI:** 10.3390/foods11091317

**Published:** 2022-04-30

**Authors:** Ke Wang, Lisong Huang, Yangting Xu, Baozhong Cui, Yanan Sun, Chuanyang Ran, Hongfei Fu, Xiangwei Chen, Yequn Wang, Yunyang Wang

**Affiliations:** 1College of Food Science and Engineering, Northwest A&F University, Yangling, Xianyang 712100, China; wangkejxsj@nwafu.edu.cn (K.W.); 2017013554@nwafu.edu.cn (Y.X.); 2015014870@nwafu.edu.cn (B.C.); synsyn@nwafu.edu.cn (Y.S.); 2019055072@nwafu.edu.cn (C.R.); fuhongfei@nwsuaf.edu.cn (H.F.); chenxiangwei@nwsuaf.edu.cn (X.C.); yequnw@163.com (Y.W.); 2College of Food Science and Engineering, NanJing University of Finance &Economics, Nanjing 210023, China; 1120211124@stu.edu.nufe.cn

**Keywords:** RF sterilization, beef sausage, dielectric properties, heating uniformity

## Abstract

Radio frequency (RF) heating has the advantages of a much faster heating rate as well as the great potential for sterilization of food compared to traditional thermal sterilization. A new kettle was designed for sterilization experiments applying RF energy (27.12 MHz, 6 kW). In this research, beef sausages were pasteurized by RF heating alone, the dielectric properties (DPs) of which were determined, and heating uniformity and heating rate were evaluated under different conditions. The results indicate that the DPs of samples were significantly influenced (*p* < 0.01) by the temperature and frequency. The electrode gap, sample height and NaCl content had significant effects (*p* < 0.01) on the heating uniformity when using RF energy alone. The best heating uniformity was obtained under an electrode gap of 180 mm, a sample height of 80 mm and NaCl content of 3%. The cold points and hot spots were located at the edge of the upper section and geometric center of the sample, respectively. This study reveals the great potential in solid food for pasteurization using RF energy alone. Future studies should focus on sterilization applying RF energy and SW simultaneously using the newly designed kettle.

## 1. Introduction

Nowadays, people’s increasing awareness of the convenience and health aspects of ready-to-eat foods has stimulated the demand for more organic and natural foods, thus promoting the development of a sausage food free of artificial additives [1,2]. However, a majority of the manufactured beef products continue to be periodically contaminated with micro-organisms like *Escherichia coli* (*E. coli*) and *Salmonella* [3,4].

The traditional process of sterilization often has a harmful impact on product quality as the result of the slow heating rate and non-uniform heating [5,6,7,8]. Some novel sterilization technologies have proven to be an effective replacement for conventional methods, including infrared, ultrasonic, ohmic heating, microwave (MW) and radio frequency (RF) technologies [9,10,11,12,13].

RF heating processing, an emerging technology for pasteurization, has been carried out as an effective method for microbial inactivation in different food materials [14]. Its function depends on the heat energy transformed by electric power as a consequence of charged ions or polar molecules migrating and rotating under radiofrequency electromagnetic fields (3000 Hz−300 MHz) [15,16]. Therefore, RF heating technology has the advantages of higher penetration depth and faster heating rate, resulting in much less processing time to complete the pasteurization process compared to conventional methods [17,18].

DPs composed of dielectric constant (*ε*′) and dielectric loss factor (*ε*″) are important factors affecting heating uniformity and heating rate [19]. To achieve a better understanding of the RF heating of beef sausage, it is essential to measure and evaluate the DPs of beef sausage. In previous research, the effects of temperature (T) and frequency (*f*) on dielectric properties of different kinds of meat and meat products have been analyzed, including chicken meat [20,21] and meat batters [22]. In addition, research on dielectric properties involving beef meat blends [23] and beef biceps femoris muscle [24], whose DPs are similar, have been reported. However, the DPs of beef sausage have been rarely reported, and further study of DPs is needed for pilot-scale RF heating.

Non-uniform heating is an important factor restricting the large-scale application of RF pasteurization. Therefore, it is necessary to study the uniformity of RF heating. Several experimental investigations on heating uniformity have been reported using RF energy. Li et al. [25] evaluated the effects on frozen beef of the heating uniformity in RF systems, observing that the RF tempering rate with immersion in glycerol solution was much higher than the traditional refrigeration tempering method. Dong et al. [26] revealed that added water and fat had negative effects on the RF heating uniformity, whereas high salt content had a positive influence on the RF heating uniformity. In addition, the uniformity index (*λ*) has been applied to analyze the heating uniformity of food, and numerous studies have been carried out on RF treatments, including those involving walnut kernels [27] and peanuts [28]. However, there is limited literature on heating uniformity in meat products during RF heating in terms of the uniformity index (*λ*).

The objectives of this research were (1) to study the DPs of beef sausage as influenced by temperature and frequency and analyze the influence of DPs on temperature at RF frequency compared to MW frequency; (2) to assess the effects of electrode gap, sample height and NaCl content in terms of the heating profile and temperature distribution of beef sausage by RF heating; (3) to evaluate the heating uniformity on the basis of the uniformity index (*λ*) when subjected to RF energy.

## 2. Materials and Methods

### 2.1. Beef Sausage Manufacturing

Qin-chuan beef (hind leg), as the main raw material of beef sausage, was obtained from a local store. Sausages were prepared with fresh beef (250 ± 1 g), ginger powder (5.0 ± 0.1 g), green Chinese onion (5.0 ± 0.1 g) and carrot powder (5.0 ± 0.1 g), minced through a high-speed blender (SP903, Supor Co., Ltd., Zhejiang, China), with the following additives: wheat starch (30.0 ± 1 g), egg (60.0 g ± 1 g) and soy sauce (5.0 ± 0.1 g). After packaging and steaming, the sample was cut into a cylindrical shape. For RF heating experiments, the beef sausages were prepared with dimensions (diameter × height) of 26 × 75 mm, 26 × 80 mm and 26 × 85 mm with mean ± SD weights of 38.8 ± 0.4 g, 40.5 ± 0.5 g and 41.8 ± 0.5 g, respectively. For DPs measurement, the samples were prepared with dimensions of 21.5 mm × 45 mm and weighed 25.5 ± 0.3 g. The samples were prepared with different NaCl content (3.0%, 2.0% and 1.0%). All batches of sausages were manufactured at a temperature of 25 °C within 1 h and then packaged and stored in a refrigerator at 4 °C. To keep the same initial temperature, the samples were transferred to an incubator at 25 ± 1 °C for 1–2 h before the experiments.

### 2.2. Physiochemical Composition of Beef Sausage

The moisture of beef sausage was measured according to the AOAC (1985) [29] standard methods 985.14. The total nitrogen content of beef sausage was measured using the Kjeldahl method following the method number 992.15 (AOAC, 1993) [30]. Crude protein was estimated by multiplying total nitrogen content by a factor of 6.25. Moreover, fat content of the sample was determined by following the method of Soxhlet extraction according to AOAC 991.36 [31]. The ash and starch of the sample were analyzed according to AOAC 923.03 [32] and AOAC 958.06 [33], respectively. The composition of the sample is summarized in Table 1. Each experiment was replicated three times.

### 2.3. Determination of DPs

The open-ended coaxial probe method is commonly used in the field of food research for measuring the DPs of different kinds of foods [34]. The method described above was chosen to measure the DPs of beef sausage in this study. The measuring system mainly includes an E4991B-300 model impedance analyzer (Keysight Technologies Inc., Santa Rosa, CA, USA), a calibration kit (E4991B-010), a SST-20 oil circulated oil bath (Wuxi Guanya Refrigeration Technology Co., Ltd., Jiangsu, China) and a high-temperature coaxial cable and dielectric probe (85070E-20). A more detailed description can be found in Cui et al. [14].

A 30–45 min warm-up and calibration of the impedance analyzer were performed prior to each test. The E4991B calibration kit was used to calibrate the impedance analyzer, with an Open, Short, and 50 Ω resistance in order. The coaxial probe was calibrated by air at first, then Short, and finally deionized water (25 °C) [17]. After calibration, the sample was put into a cylindrical sample holder with dimensions of 21.5 mm × 45 mm, which was surrounded by a circulating temperature-controlled oil bath. A pre-calibrated type-T thermocouple temperature sensor was applied to test the central temperature of the sample. The sample was kept in close contact with the dielectric probe and sealed in the test cell. The temperature of the three samples was controlled from 25 to 90 °C at an interval of 10 °C (25, 30, 40, 50, 60, 70, 80 and 90 °C), with frequency range between 1 MHz and 3000 MHz. This method was similar to studies on lean beef meats [35] and meat lasagna [36]. Each treatment was replicated three times.

### 2.4. RF Heating System

A 6 KW, 27.12 MHz pilot-scale free-running oscillator RF heating system (GJG-2.1-10A-JY; Hebei Huashijiyuan High Frequency Equipment Co., Ltd., Shijiazhuang, China) was used in this research. This system was composed of an RF feeder, four inductors and two electrode plates. Different electrode gaps between 100–300 mm could be obtained by adjusting the top electrode plate. Detailed descriptions of the RF system can be found in Cui et al. [14]. The RF heating system was equipped with two parts of a sterilization kettle and a fluorescence optical fiber temperature measurement system (Figure 1). A customized sterilization kettle was designed, consisting of two lid plates (Ø 280 mm × 20 mm), a hollow cylindrical vessel (Ø 280 mm × 110 mm) made of PTFE, and a flange (Figure 2). For sealing, the top and bottom lid, which were both made of aluminum alloy, were fixed with to vessel by screws. The real-time changes in the temperature of samples was tested through a fluorescence-based optical fiber temperature measure system (HQ-FTS-D1F00, Xi’an Heqi Opto-Electronic Technology Co., Ltd., Xi’an, China) in the processing of RF heating. The sterilization kettle provided a relatively thermal insulation environment for materials, and a sterilization kettle using RF energy alone was selected in this study.

### 2.5. Evaluating the Heating Uniformity and Heating Rate

Prior to starting this experiment, the cylinder-shape sample (Ø 26 mm × 80 mm) was filled in a flat-bottom tube made of PTFE, and the tube filled with sample was placed on the geometric center of the bottom lid in the sterilization kettle (Figure 2). Three fiber optic sensors were inserted at positions 1, 2 and 3 in the sample to test the internal temperature during RF heating processing (Figure 3a). Then, the top lid was tightened with bolts. The sterilization kettle was placed on the bottom electrode in the RF cavity (Figure 1).

After RF heating experiments, the sample temperature distribution was determined instantly using the infrared thermography camera (FLIR A300, FLIR Systems Inc., Wilsonville, OR, USA). Individual surface temperature data points of 2500–10,000 for each thermal image were recorded and collected using the software (BM_IR V7.4).

#### 2.5.1. Electrode Gap

Three electrode gaps of 175, 180 and 185 mm were selected according to the arrangement of the experiments. Then, the heating system of the RF was opened. The time-temperature profile was recorded to find the point with the slowest heating rate, which was considered to be the cold point. The RF heating system was turned off when the temperature at the cold spot reached 70 °C [37,38,39] for pasteurization. After RF heating was turned off, the sample was removed from the tube carefully and quickly. The sample was photographed using the infrared thermography camera to obtain thermal images of upper (A), middle (B) and lower layers (C), as well as the longitudinal section (D) (Figure 3). The whole process lasted less than 30 s from turning off the RF system to the end of photographing.

#### 2.5.2. Sample Height

Samples with different heights (75, 80 and 85 mm) were selected in the processing of RF heating. The electrode gap of 120 mm was used based on previous experiments. The PTFE tube filled with the sample was placed on the geometric center of the bottom lid in the sterilization kettle (Figure 2). Three fiber optic sensors were inserted at positions 1, 2 and 3 of the beef sausage to test the internal temperature during RF heating, as illustrated in Figure 3a. The RF heating system was turned off when the temperature at the cold spot reached 70 °C [37]. The temperatures of the upper, middle and lower layer and the longitudinal section were analyzed and mapped as mentioned previously.

#### 2.5.3. NaCl Content

Based on previous experiments, the determined electrode gap and sample height were 180 mm and 80 mm, respectively. Sausage samples were prepared with different NaCl content (3.0%, 2.0% and 1.0%) for RF heating. The sample and the fiber optic sensors were placed as described in Section 2.5.1 and Section 2.5.2. At the slowest heating point, which was considered to be the cold point reaching the target temperature (70 °C) for pasteurization, the RF system was turned off. When RF heating was stopped, the temperatures of the upper, middle and lower layers, as well as the longitudinal section, were analyzed using the same method as described above.

### 2.6. Assessment of the Heating Uniformity

The heating uniformity was evaluated by the thermal images captured by the infrared camera, and the heating uniformity index *λ* was calculated using Equation (1) [13,16,40]
(1)$$\lambda \;=\frac{\sqrt{\sigma^{2}-\sigma_{0}^{2}}}{\mu -\mu_{0}}$$
where *μ*_0_ and *μ* are the initial and final sample temperatures, and *σ* and *σ*_0_ are the final and initial SDs of sample temperatures in the RF process, respectively.

### 2.7. Statistical Analysis

Three trials were conducted to obtain mean values and standard error for statistical analysis. The contour plots of data were created by Origin 2020 (Origin Lab Corp., Northampton, MA, USA) to analyze the temperature distribution of different layers of the sample, and the average temperature as well as the SDs of each layer were obtained using Microsoft Excel (Microsoft Office, Redmond, WA, USA). Microsoft Excel was also used to analyze the variance (ANOVA), and the Tukey test with a confidence level of 99% was performed in SPSS version 16.0 (SPSS, Chicago, IL, USA).

## 3. Results and Discussion

### 3.1. DPs of Beef Sausage

Figure 4 indicates the DPs of beef sausages as a function of frequency (*f*) at different temperatures. Both dielectric constant (*ε*′) and loss factor (*ε*″) values decreased as the frequency increased in the samples. Similar trends were observed in the dielectric properties of noodles, beef meatballs and sauce [36]. At a frequency lower than 300 MHz, *ε*′ significant increased (*p* < 0.01) at temperatures ranging from 25 to 90 °C, while it increased slightly at 300 MHz or higher as the test temperature increased. The effects of frequency on *ε*″ showed a different trend with *ε*′. *ε*″ decreased with the decrease of temperature among the range of frequencies from 13.56 MHz to 2450 MHz. This was probably because of free water dispersion and ionic conduction, which govern the change of *ε*″ [41]. At a frequency higher than 2450 MHz, both *ε*′ and *ε*″ varied within a narrow range.

The measured DPs of the sample as a function of temperature are presented in Figure 5. The dielectric constant at three RF frequencies (13.56, 27.12 and 40.68 MHz) are higher than those at microwave frequencies (915 and 2450 MHz), and this increased with an increasing temperature as compared with no significant changes at microwave frequencies. Similar results have also been reported in meat batters [22]. The denaturation of protein at high temperature will cause the release and shrinkage of water, which is considered the cause of the significant changes in the dielectric properties of beef sausage. The loss factor of beef sausages at RF frequencies is much higher than that at microwave frequencies, and this increased with an increasing temperature both in RF and MW. Similar tendencies were also seen in previous research [21,22]. Providing a uniform and stable electric field is significantly important for pilot-scale RF pasteurization at 27.12 MHz [42].

Two predictive models of *ε*′ and *ε*″ were created as a function of temperature (T) and frequency (*f*) by nonlinear regression analysis in beef sausage samples in Equations (2) and (3).

Both temperature (*T*) and frequency (*f*) had significant effects on (*ε*′) and (*ε*″), and the R^2^ values were 0.941 and 0.951, respectively (*p* < 0.001).
(2)$$\varepsilon^{\prime }=\;183.937-\;3.129f+0.160T\;+\;{0.004f}^{\;2}+{0.006T}^{2}-\;0.01\mathrm{fT}\;-\;{1.316\;\times \;10}^{-6}f^{\;3}-\;{1.323\;\times \;10}^{-5}T^{3}+{4.171\;\times \;10}^{-7}{\mathrm{Tf}}^{\;2}-\;{1.758\;\times \;10}^{-6}{f\;T}^{\;2}\;$$
(3)$$\varepsilon^{\prime \prime }=2925.437\;-\;90.397f\;+\;28.767T+{0.131f}^{\;2}+{0.024T}^{2}-\;0.043\mathrm{fT}-\;{3.84\;\times \;10}^{-5}f^{\;3}+{3.797\;\times \;10}^{-4}T^{3}+{1.345\;\times \;10}^{-5}{\mathrm{Tf}}^{\;2}-\;{1.837\;\times \;10}^{-5}{f\;T}^{\;2\;}$$

The regression models are suited for predicting the DPs of beef sausage under a temperature of 25 to 90 °C and at the frequency of 13.56 to 2450 MHz. Further research should focus on the DPs of the different moisture content and containers of beef sausage, using computer simulation system to improve RF heating uniformity.

The DPs of samples with different NaCl contents were measured at 27.12 MHz and are shown in Table 2. The results revealed that *ε*′ increased with the increase of NaCl content, and a similar trend was obtained with *ε*″.

### 3.2. Determination of Cold Spot of RF Heating

Testing the cold spots during the RF heating process is crucial for pasteurization and affects the processing technology, safety, quality, and cost of food materials [43,44,45]. According to the description above, three electrode gaps (175, 180 and 185 mm), sample heights (75, 80 and 85 mm), and NaCl contents (1%, 2%, and 3%) were selected. The results of the temperature distribution are shown in Table 3. Within the same period of time, the temperature of the upper layer in Figure 3a was much lower that of the middle layer and lower layer when subjected to RF heating. The table also showed that the temperature of the middle part was slightly higher than that of the lower part, which meant the heating hot spot was generally near the geometric center. A similar phenomenon was reported for potato cuboids [16], which was attributed to the electric field behavior and heat loss at the sample edge. Therefore, the sample position 1 was selected as the cold spot for subsequent study.

### 3.3. Evaluating the Heating Uniformity and Heating Rate in the Different Conditions

#### 3.3.1. Electrode Gap

Figure 6 reveals the heating profile of time–temperature and contour plots of samples during the RF treatment with different electrode gaps. The RF heating rates increased as the electrode gap decreased (Figure 6a), and the maximum heating rate was obtained under an electrode gap of 175 mm. This phenomenon may be due to the distance between the sample and upper electrode plate. The sample could absorb more electric energy when the distance was closer [46]. Similar trends were reported for frozen chicken breast meat [47] and frozen tilapia fillets [48]. The detailed temperature distribution of the beef sausage at the upper, middle, lower and longitudinal sections is shown in Figure 6b when the RF system was shutting down. The results in this figure indicated that the cold spots were at the edges, while the hot spots were at the center of the beef sausage, and the temperature close to the top was much lower than temperature near the middle and bottom, which was consistent with the results in Table 3. It also can be seen from the contour plots that the temperature distribution was more uniform at an electrode gap of 180 mm as compared to the others. The temperatures of upper parts were below 70 °C because of the heat dissipation when the beef sausage samples were removed from the sterilization kettle.

Table 4 shows the heating uniformity index (*λ*) values of beef sausage samples in the upper, middle and lower layers and the longitudinal section at different NaCl (1, 2 and 3%) content with different sample heights (75, 80 and 85 mm) under different electrode gaps (175, 180 and 185 mm) at the initial temperature of 25 °C when applied to RF heating. The results showed that there were no significant effects (*p* > 0.01) on the electrode gaps and layers, and the lowest uniformity index was obtained under the electrode gap of 180 mm. High heating rates corresponding to high throughputs may have caused the non-heating uniformity due to rapid and runaway heating [49,50]. Thus, the electrode gap of 180 mm was used for additional research.

#### 3.3.2. Sample Height

Figure 7 shows the profiles of the time–temperature and contour distribution of samples when subjected to RF energy with three different sample heights (75 mm, 80 mm and 85 mm). The RF heating rates increased with an increasing sample height when the electrode gap was 180 mm and NaCl content was 3%. Li et al. (2018) obtained similar trends of the heating rate increasing as the food thickness increases from 4 cm and 5 cm to 6 cm of frozen beef samples with various thicknesses under RF treatment [46]. The contour plots in Figure 7b show the temperature distribution of beef sausage with different sample heights after RF heating. It can be observed in the contour plots that temperature distribution was more uniform when the sample height was 80 mm. At the same time, the upper layer was more uniform according to the results of temperature distribution in general. The highest temperatures were obtained at the middle layer compared to those of upper and lower layers. Table 4 compares the heating uniformity index (*λ*) of three sample heights after RF heating when all beef sausage sample temperatures of cold spots reached 70 °C. The lowest *λ* of the sample was obtained with a height of 80 mm. This was because the deflection of the electric field increased the electric field intensity at the geometric center of the sample [51]. A sample of 80 mm had the fastest heating rate and the lowest heating uniformity index when all beef sausage sample temperatures of cold spots reached 70 °C. Based on the above results, a fixed sample height of 80 mm was selected for further research.

#### 3.3.3. NaCl Content

Figure 8 demonstrates the time–temperature profile and contour plots of temperature distribution of samples with different NaCl contents by RF heating. Figure 8a shows that the slowest heating rate was achieved when the NaCl content was 1%. The heating rate increased with the increase of NaCl content at first. However heating rates slowed down when NaCl content was more than 2%, which showed a similar trend to the study of Jeong et al. [52]. This phenomenon may be due to the effect of salty shield, which leads to the slowdown of the heating rate [53]. It revealed that the temperature distribution was more uniform when NaCl contents were 1% and 3%, as seen in Figure 8b. The minimum *λ* value was obtained with the NaCl content of 3%, as seen in Table 4. Dong et al. (2021) also reported that RF heating uniformity might be improved with the high salt concentrations of ground beef [26]. In addition, the sample with a higher salt content could provide a better value of sensory evaluation and preservation characteristics. Based on the above results, a sample with 3% NaCl content had the fastest heating rate and the lowest heating uniformity index. Therefore, the sample with the NaCl content of 3% was selected for future studies.

## 4. Conclusions

A customized sterilization kettle was designed for RF pasteurization of beef sausage. The sterilization kettle was well applied in an RF heating system according to efficiency and stability tests. The temperature distribution indicated that the cold points were at the upper layer and edge of the sample, whereas the hot spots were at the geometric center. Temperature and frequency had significant influence (*p* < 0.01) on the DPs of the beef sausage sample. The best heating uniformity was obtained at an electrode gap of 180 mm, sample height of 80 mm and NaCl content of 3%. Overall, this research enriches the information on the heating uniformity of foods for pasteurization under RF systems. Additional studies should focus on the quality effects and pasteurization kinetics of food. Moreover, the sterilization system described above can also realize RF combined with super high temperature water (RFSW) sterilization for subsequent tests in the future.

## Figures and Tables

**Figure 1 foods-11-01317-f001:**
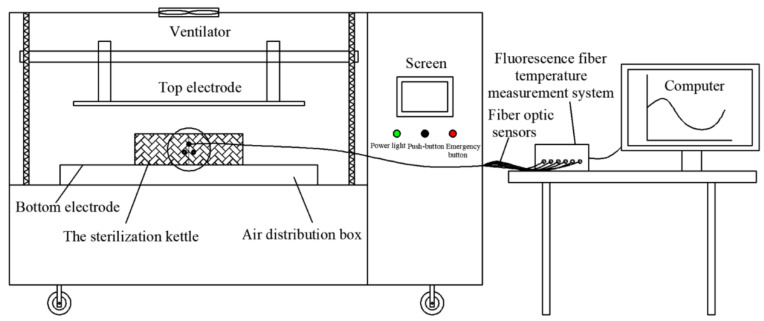
Simplified schematic diagram of sterilization kettle in the RF heating system.

**Figure 2 foods-11-01317-f002:**
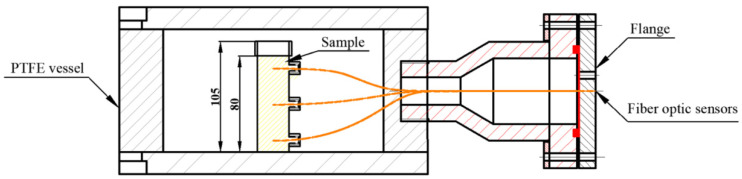
Detailed cross-section view of the sterilization kettle with beef sausage placed in the center and fiber optic sensor inserted in the sample.

**Figure 3 foods-11-01317-f003:**
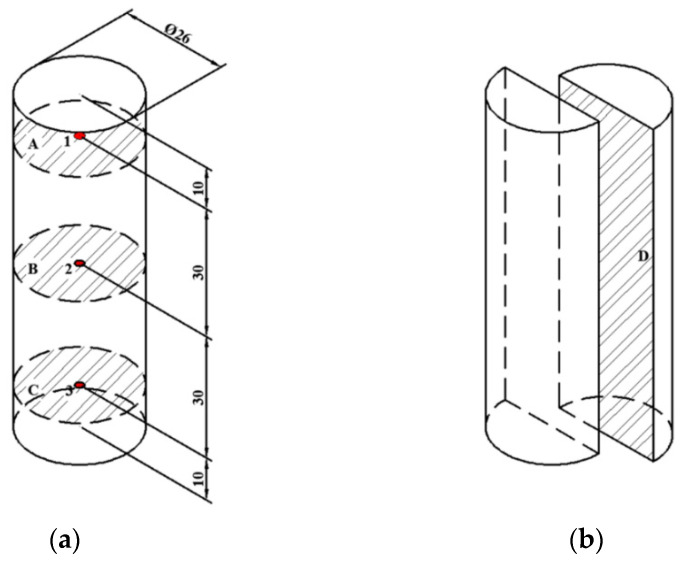
The location of fiber optic sensors (1, 2, 3) and four layers (upper, A; middle, B; lower, C; longitudinal, D) in the beef sausage sample for temperature distribution measurement (all measured in millimeters). (**a**) Detailed diagram of sample. (**b**) Longitudinal section of sample.

**Figure 4 foods-11-01317-f004:**
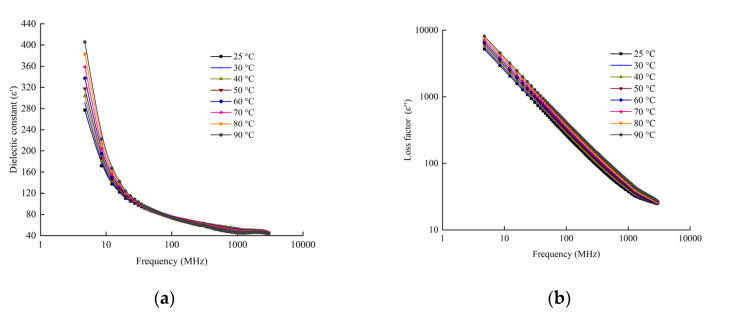
DPs related with frequency of beef sausage at different temperatures (25, 30, 40, 50, 60, 70, 80 and 90 °C): (**a**) dielectric constant (*ε*′) as dependent variable; (**b**) dielectric loss factor (*ε*″) as dependent variable.

**Figure 5 foods-11-01317-f005:**
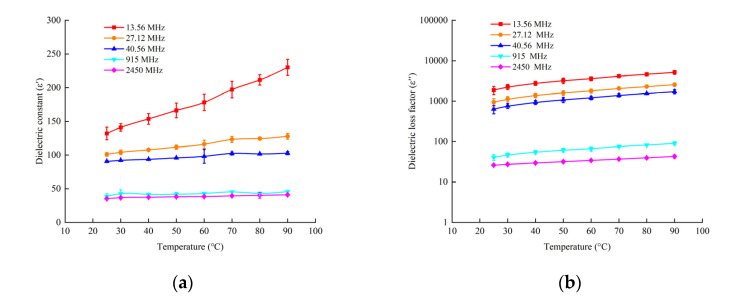
Temperature-dependent DPs of samples at five selected frequencies: (**a**) dielectric constant (*ε*′) as dependent variable; (**b**) dielectric loss factor as dependent variable. RF band included 13.56, 27.12 and 40.68 MHz; microwave frequency included 915 and 2450 MHz.

**Figure 6 foods-11-01317-f006:**
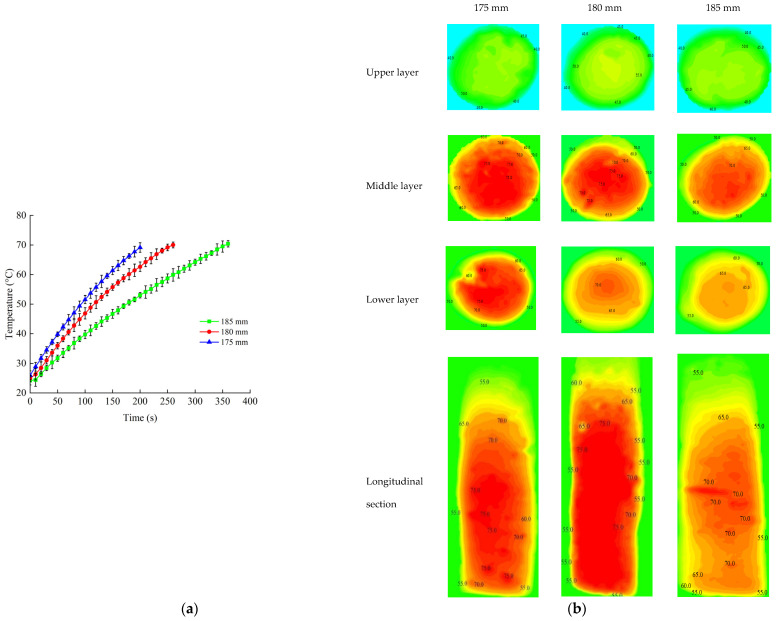
Time–temperature heating curves of beef sausages (**a**) and the contour plots of temperature distribution (**b**) at different electrode gaps (175, 180 and 185 mm) at a sample height of 80 mm and when the temperature was initialized at 25 °C.

**Figure 7 foods-11-01317-f007:**
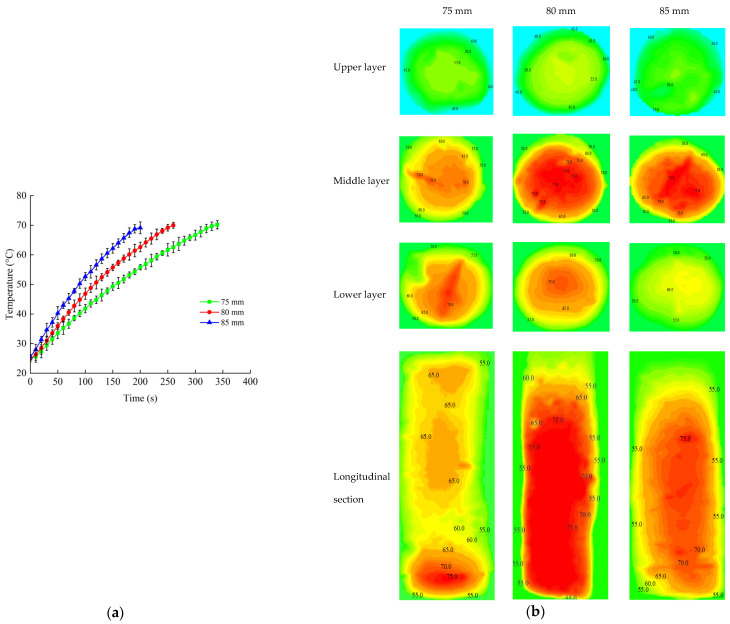
Time–temperature heating curves of beef sausages (**a**) and the contour plots of temperature distribution (**b**) at different sample heights (75, 80 and 85 mm) under an electrode gap of 180 mm and when the temperature was initialized at 25 °C.

**Figure 8 foods-11-01317-f008:**
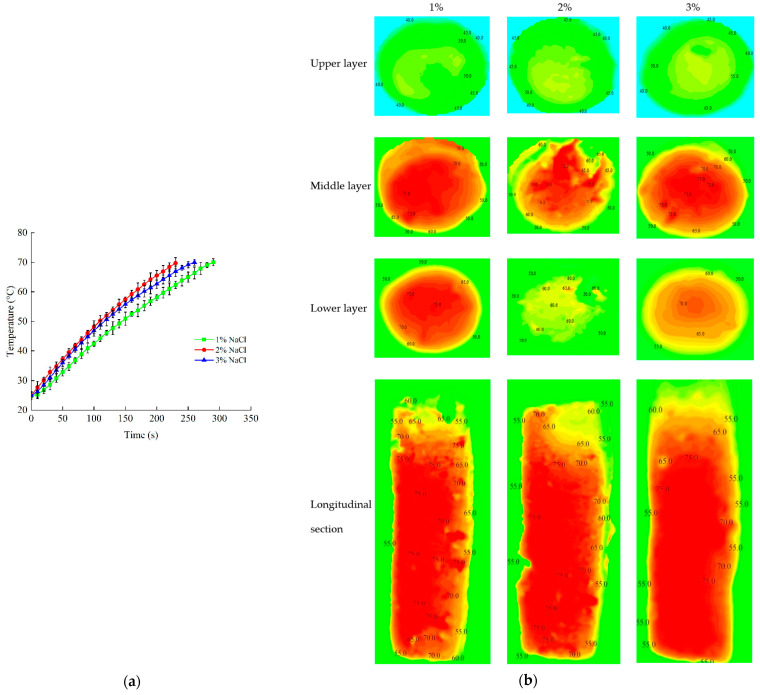
Time–temperature heating curves of samples (**a**) and the contour plots of temperature distribution (**b**) with different NaCl contents (1%, 2% and 3%) at an electrode gap of 180 mm, a sample height of 80 mm and when the temperature was initialized at 25 °C.

**Table 1 foods-11-01317-t001:** The composition of beef sausage.

Moisture Content (g kg^−1^ w.b.)	Ash (g kg^−1^)	Protein (g kg^−1^)	Carbohydrate (g kg^−1^)	Fat (g kg^−1^)
633.5	22.5	171.1	122.5	38.9

**Table 2 foods-11-01317-t002:** The DPs of samples with different NaCl contents (1%, 2% and 3%) at an RF frequency of 27.12 MHz and a temperature of 25 °C.

Frequency (MHz)	NaCl Content (%)	Dielectric Properties	
		*ε*′ ± SD	*ε*″ ± SD
27.12	1	96.69 ± 2.28	676.52 ± 14.57
	2	98.95 ± 2.01	806.92 ± 16.99
	3	101.01 ± 2.83	937.28 ± 21.76

**Table 3 foods-11-01317-t003:** The temperature of different layers of samples with different NaCl contents (1, 2 and 3%) under different sample heights (75, 80 and 85 mm) at different electrode gaps (175, 180 and 185 mm) when the temperature of the cold spot reached 70 °C and the temperature was initialized as 25 °C.

Electrode Gap (mm)	Sample Height (mm)	NaCl (%)	Layer		
			Upper	Middle	Lower
175	80	3	70.7 ± 0.7	79.9 ± 1.1	69.0 ± 0.9
180	80	3	70.3 ± 0.4	93.2 ± 1.6	99.4 ± 1.1
185	80	3	70.4 ± 0.6	83.9 ± 2.2	87 ± 2.3
180	75	3	70.4 ± 0.6	90.6 ± 0.7	78.3 ± 0.6
180	80	3	70.3 ± 0.4	93.2 ± 1.6	99.4 ± 1.1
180	85	3	70.0 ± 1.2	96.8 ± 0.9	94.7 ± 1.3
180	80	1	70.2 ± 0.5	91.3 ± 0.8	81.3 ± 0.9
180	80	2	70.1 ± 0.7	95.5 ± 1.2	102.4 ± 1.4
180	80	3	70.3 ± 0.4	93.2 ± 1.6	99.4 ± 1.1

**Table 4 foods-11-01317-t004:** Beef sausage *λ* value of different layers at different NaCl contents (1, 2 and 3%) with different sample heights (75, 80 and 85 mm) under different electrode gaps (175, 180 and 185 mm) at the initial temperature of 25 °C.

Electrode Gap (mm)	Sample Height (mm)	NaCl (%)	*λ*			
			Upper	Middle	Lower	Longitudinal
175	80	3	0.341 ± 0.0025 ^a^	0.343 ± 0.0044 ^a^	0.464 ± 0.0054 ^d^	0.455 ± 0.0039 ^cd^
180	80	3	0.335 ± 0.0016 ^a^	0.337 ± 0.0051 ^a^	0.406 ± 0.0026 ^b^	0.441 ± 0.0015 ^c^
185	80	3	0.404 ± 0.0054 ^b^	0.341 ± 0.0036 ^a^	0.414 ± 0.0046 ^b^	0.453 ± 0.0047 ^cd^
180	75	3	0.344 ± 0.0012 ^bc^	0.331 ± 0.0034 ^a^	0.414 ± 0.0033 ^d^	0.404 ± 0.0046 ^d^
180	80	3	0.335 ± 0.0016 ^a^	0.337 ± 0.0051 ^a^	0.406 ± 0.0026 ^b^	0.441 ± 0.0015 ^c^
180	85	3	0.351 ± 0.0041 ^c^	0.348 ± 0.0022 ^c^	0.438 ± 0.0013 ^e^	0.449 ± 0.0012 ^f^
180	80	1	0.366 ± 0.0011 ^b^	0.344 ± 0.0022 ^a^	0.445 ± 0.0031 ^e^	0.475 ± 0.0016 ^f^
180	80	2	0.378 ± 0.0034 ^c^	0.359 ± 0.0041 ^b^	0.438 ± 0.0039 ^e^	0.467 ± 0.0021 ^f^
180	80	3	0.335 ± 0.0016 ^a^	0.337 ± 0.0017 ^a^	0.406 ± 0.0021 ^d^	0.441 ± 0.0023 ^e^

Different lowercase letters indicate a significant difference (*p* < 0.01).

## Data Availability

Data on this study are available in the article.

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
