# Peer review of "Evaluation of Pilot-Scale Radio Frequency Heating Uniformity for Beef Sausage Pasteurization Process"

_foods, 2022, doi:10.3390/foods11091317_

Round 1
Reviewer 1 Report
12- You mention that you use superheated water,but in the next line you claim that you are going to use RF only. Superheated is only used in the abstract. Have you forgot to describe some of the equipment you used. Please clarify.
16- You indicate that the results are influenced by the frequency, but you mention that you are using RF at 27.12 MHz.
66- Is it 1MHz-3GHz or 1 GHz-3 GHz. If you are referring to first case place clarify because it can be confusing.
77- Was the product stored for stabilisation, etc?
95- Pls correct the caption of the table
119- What is majorly?
Fig 1 and 2. It is not clear how the left part of the kettle (Fig 2) fits in the kettle represented in Fig 1.
I think methodology requires a lot of inprovement. Things should be much clearer explained and the information reorganized. For example, lines 161-164 may go to section 2.7.
Section 2.5.1 Is all the information referring to the electrode gap? I think new sections are needed to make more undestandable the description of the methodology. Similar for 2.5.2
In section 2.5.3, an electrode gap of 120mm is selected. However in section 2.5.1 only electrode gaps of 170-175-180mm are studied.
If you are focusing your study on RF, maybe Figure 4 are not necessary and you should only focus on the results in the RF frequencies.
Please discuss the reason for the obtained results and the implication of these results. You are just describing the results you obtained. You should as well as discuss your results more in depth in the context of similar works.
271-272 I do not think that results obtained in almonds
Author Response
Response to Reviewer 1 Comments
Dear,
Thank you for your careful reconsideration of our manuscript. I really appreciate all the comments and suggestions! We have studied the comments carefully and attempted to address all of the raised issues. These comments are helpful for improving our paper. The response to the comments are as following.
Point 1: 12-You mention that you use superheated water, but in the next line you claim that you are going to use RF only. Superheated is only used in the abstract. Have you forgot to describe some of the equipment you used. Please clarify.
Response 1: Thank you for pointing this out. We have described the SW system in more detail in 2.4 ( line 126-138)
Point 2: 16- You indicate that the results are influenced by the frequency, but you mention that you are using RF at 27.12 MHz
Response 2: The results in this article indicate that the DPs of samples were significantly influenced by the temperature and frequency, which is important for dielectric heating especially for radio frequence or microwave. Only in this research, a 6 KW, 27.12 MHz RF heating system was operated in this research for pilot-scale testing. The sample with different treatments (frequency from 1MHz to 3GHz) could be discessed as a comparison for the further study.
Point 3: 66- Is it 1MHz-3GHz or 1 GHz-3 GHz. If you are referring to first case place clarify because it can be confusing. (1MHz-3GHz)
Response 3: Thank you for your suggestion. We are sorry for our incorrect writing, we have modified the expression into “(1MHz-3GHz)” in line 66.
Point 4: 77-Was the product stored for stabilisation, etc?
Response 4: Yes, we hope that the properties of the samples could remain relatively stable before the test, so that we could obtain the results of higher repeatability.
Point 5: 95-Pls correct the caption of the table
Response 5: Thank you for your suggestion. Sorry for the mistake, and we have added correct title “ The compositions of beef sausage“ at line 95.
Point 6: 119- What is majorly?
Response 6: We just describe the main components of RF heating system, because they are the core parts in the process of test operation. The rests, such as bolts and screws, have no special role and are not mentioned here.
Point 7: Fig 1 and 2. It is not clear how the left part of the kettle (Fig 2) fits in the kettle represented in Fig 1.
Response 7: Thank you for your suggestion. We have added the more details in section 2.4 (line 126-133). Combined the descripion above with section 2.5 (line 151-156), we think it is much clear about the issue you mentioned.
Point 8: I think methodology requires a lot of inprovement. Things should be much clearer explained and the information reorganized. For example, lines 161-164 may go to section 2.7.
Response 8: Thank you for your suggestion. We have deleted this part, and added the all the issues about data analysis into section 2.7 (line 199-204).
Point 9: Section 2.5.1 Is all the information referring to the electrode gap? I think new sections are needed to make more undestandable the description of the methodology. Similar for 2.5.2
Response 9: Thank you for your comments. We have regrouped the paragraph in section 2.5 (line 151-156) to make it more clear and undestandable.
Point 10: In section 2.5.3, an electrode gap of 120mm is selected. However in section 2.5.1 only electrode gaps of 170-175-180mm are studied.
Response 10: Thank you for your suggestion. Sorry for the error of writing. We have modified the it in section 2.5.3 (line 184).
Point 11: If you are focusing your study on RF, maybe Figure 4 are not necessary and you should only focus on the results in the RF frequencies.
Response 11: Thank you for your suggestion. The results in this article indicate that the DPs of samples were significantly influenced by the temperature and frequency, which is important for dielectric heating especially for radio frequence or microwave. Only in this research, a 6 KW, 27.12 MHz RF heating system was operated in this research for pilot-scale testing. We think it is necessary, mainly because studying the DPs of beef sausage at frequencies from 1MHz to 3GHz can provide theoretical supports for the same type of food in the field of dielectric heating (RF, microwave, etc.)
Point 12: Please discuss the reason for the obtained results and the implication of these results. You are just describing the results you obtained. You should as well as discuss your results more in depth in the context of similar works.
Response 12: Thank you for pointing this out. We have modified the some discussions in line 267,287,291 and 297,and added more discussions in line 287-289 and 267-268.
Point 13: 271-272 I do not think that results obtained in almonds
Response 13: Thank you for your reminder. We have deleted it and changed a new discussion according to resulits of apple slice RF blanching in line288.
Reviewer 2 Report
Manuscript is well written and systematic. There is need to input some gaps between lines in case of table titles etc. (no exact comments from my side). I have only minor comments given in the following list.
- Abbreviation DP is explained first at line 43 but it is used at lines 14 and 15. It has to be explained at first occasion.
- Line 95 contains mistaken text “This is a table. Tables should be placed in the main text near to the first time they are cited.” Please input here correct title of the Table 1.
- Lines 106 and 107 contain sentence “. The coaxial probe was calibrated by air, short and deionized water (25 °C) in order [17]. “End of sentence gives me no sense “in order [17]“.
- Line 124 remove “s” from word “lids”.
- Line 144 symbol for diameter is not correct. It is a Greek letter phi. True symbol for diameter is circle crossed out by a slash “∅”. This symbol has number 198 in a word writer.
- Line 187 contains sentence “Where μ and μ0 are initial and final sample temperatures”. I think that μ is final sample temperature. Logically the same correctly stated is declared in following sentence about sigma’s.
- Line 206 contains declaration “The DPs of sample as a function of temperature are measured and shown in Figure 5.” I do not believe you that figure measured. Please change the sentence in sense like “The measured DPs of sample as a function of temperature are presented in Figure 5”.
- Figures 6a, 7a and 8a: I do not exactly understood where were these temperatures measured or whether these temperatures are mean values of temperatures measured at points 1, 2 and 3.
Author Response
Response to Reviewer 2 Comments
Dear,
Thank you for your careful reconsideration of our manuscript. I really appreciate all the comments and suggestions! We have studied the comments carefully and attempted to address all of the raised issues. These comments are helpful for improving our paper. The response to the comments are as following.
Point 1: Abbreviation DP is explained first at line 43 but it is used at lines 14 and 15. It has to be explained at first occasion.
Response 1: Thank you for your kind reminder. We have added the explaination of DPs ( line 14 ) when it first appeared in this article, and deleted the needless explaination of DPs at line 43.
Point 2: Line 95 contains mistaken text “This is a table. Tables should be placed in the main text near to the first time they are cited.” Please input here correct title of the Table 1.
Response 2: Thank you for your suggestion. Sorry for the mistake, and we have added correct title “ The compositions of beef sausage“ at line 95.
Point 3: Lines 106 and 107 contain sentence “. The coaxial probe was calibrated by air, short and deionized water (25 °C) in order [17]. “End of sentence gives me no sense “in order [17]“.
Response 3: Thank you for pointing this out. We have changed the sentence into “The coaxial probe was calibrated by air at first, then Short , and deionized water (25 °C) at last. “ at line 107
Point 4: Line 124 remove “s” from word “lids”.
Response 4: Thank you for your suggestion. We have corrected it in line 125.
Point 5: Line 144 symbol for diameter is not correct. It is a Greek letter phi. True symbol for diameter is circle crossed out by a slash “∅”. This symbol has n umber 198 in a word writer..
Response 5: Thank you for your suggestion, we have modified this symbol into “∅ “ at line 151.
Point 6: Line 187 contains sentence “Where μ and μ0 are initial and final sample temperatures”. I think that μ is final sample temperature. Logically the same correctly stated is declared in following sentence about sigma’s..
Response 6: Thank you for your reminder. We are sorry about the wrong order of “μ0 and μ“. We have corrected it with right order at line 195.
Point 7: Line 206 contains declaration “The DPs of sample as a function of temperature are measured and shown in Figure 5.” I do not believe you that figure measured. Please change the sentence in sense like “The measured DPs of sample as a function of temperature are presented in Figure 5”.
Response 7: Thank you for pointing this out. Sorry for our incorrect writing. We have changed wrong sentence into “The measured DPs of sample as a function of temperature are presented in Figure 5“, and thanks again for mentioning the better and appropriate expression at line 218.
Point 8: Figures 6a, 7a and 8a: I do not exactly understood where were these temperatures measured or whether these temperatures are mean values of temperatures measured at points 1, 2 and 3.
Response 8: Based on our pre-experiments in 3.2 ( line 243-253 ), testing the cold spots during RF heating process is crucial for pasteurization, and the results in Table 2 shows that the temperature of upper layer in Fig 3 (a) ( point 1 ) was much lower that of the middle layer and lower layer when subjected to RF heating. Therefore, sample of point 1 was selected as cold spot for subsequent study, and figures 6a, 7a and 8a showed the time–temperature heating curves of sample at the location of point 1 in Fig 3 (a) as well.

Round 2
Reviewer 1 Report
Ln 48-50. I think you should focus on works dealing with meat products, similar to the product you are using in your investigation.
Ln 53-64. I will try to focus on analysing heating uniformety in meat products.
Ln 66. I think you should just focus on the radiofrequency domain.
Ln 115-117. I think comparison should focus on similar products.
Ln 130-132. You say that you connect the kettle to a SW equipment, but later in Ln 137-139 it seems that you do not use this system. Am I rigth? For me this quite confusing.
3.1 As I have saide before i do not see the point of including in the analysis other frequencies.
Fig.1(b) epsilon'' is usually represented in a log scale.
When evaluating the dielectric properties I think you should study the properties for all the NaCl contents.
Ln 219-230. Some of the comments are quite obvious and should be discussed in relation to other similar works (meat products)
S3.2
Ln 247. Why did you choose a NaCl content of 3%?? This decissions should be justified somewhere in the paper.
I think results should be much better discussed in relation to other similar works (meat products).
Section 3.3.1
I think results should be much better discussed in relation to other similar works (meat products). The discussion in relation to previous work is quite poor. This also applies to other results section (3.3.3 seems ok).
Are threre no conclussions of the study?
Author Response
Point 1: Ln 48-50. I think you should focus on works dealing with meat products, similar to the product you are using in your investigation.
Response 1: We have change the descriptions of previous research, focused on works about meat products including chiken meat and etc ( line 47-50)
Point 2: Ln 53-64. I will try to focus on analysing heating uniformety in meat products.
Response 2: According to your advise, we have focused on analysing heating uniformety in meat products at line 56-69.
Point 3: Ln 66. I think you should just focus on the radiofrequency domain.
Response 3: Thank you for your suggestion. We have change the description of the objectives of this study at line 71-73, and focused more on RF and relaitonship between RF and MW (line 217-221, 230,235-237).
Point 4: Ln 115-117. I think comparison should focus on similar products.
Response 4: According to your advise, we have focused on analysing heating uniformety in meat products including lean meats and meat lasagna at line 121-122
Point 5: Ln 130-132. You say that you connect the kettle to a SW equipment, but later in Ln 137-139 it seems that you do not use this system. Am I rigth? For me this quite confusing.
Response 5: After careful consideration, we have deleted the related parts of SW equipment (line 12-13, 134-137), and just retained some related prospects in abstract and conclusion.
Point 6: 3.1 As I have saide before i do not see the point of including in the analysis other frequencies.
Response 6: We have added more analysis at section 3.1 (line 217-221,249-252)
Point 7: Fig.1(b) epsilon'' is usually represented in a log scale.
Response 7: Thank you for your suggestion. We think you might mean the presentation of Fig. 4(b).
We have change it into a log scale at line 215-216.
Point 8: When evaluating the dielectric properties I think you should study the properties for all the NaCl contents.
Response 8: Thank you for your suggestion. We have added the dielectric properties of NaCl contents(1%, 2%, and 3%) in sections 3.1 (253-259)
Point 9: Ln 219-230. Some of the comments are quite obvious and should be discussed in relation to other similar works (meat products)
Response 9: Thank you for your comments. We have focused more on the research about meat products at line 230 and line 235.
Point 10: Ln 247. Why did you choose a NaCl content of 3%?? This decissions should be justified somewhere in the paper.
Response 10: Thank you for your suggestion. It is necessary to add the data of 1% and 2 % of NaCl contents.So, we have changed the description at section 3.2(line 264 and 275) and put a new Table 3 in this section.
Point 11: I think results should be much better discussed in relation to other similar works (meat products)
Response 11: Thank you for pointing this out. We have modified the some discussions in line 324-340,343-345 in section 3.3.2 and 358-369 in in section 3.3.3.
Point 12: Section 3.3.1 I think results should be much better discussed in relation to other similar works (meat products). The discussion in relation to previous work is quite poor. This also applies to other results section (3.3.3 seems ok).
Response 12: Thank you for your suggestion. We have described and discussed this part in relation works more about meat products in 3.3.1 (line 288-296, 301-303,313-315)
Point 13: Are threre no conclussions of the study?
Response 13: Thank you for your reminder. We have modified the title of section 4 at line 376
